# Risk factors for in-hospital mortality in laboratory-confirmed COVID-19 patients in the Netherlands: A competing risk survival analysis

Gerine Nijman[1,2], Maike Wientjes[3], Jordache Ramjith[4], Nico Janssen[2,5], Jacobien Hoogerwerf[1,2], Evertine Abbink[1], Marc Blaauw[6], Ton Dofferhoff[7], Marjan van Apeldoorn[8], Karin Veerman[9], Quirijn de Mast[1,2], Jaap ten Oever[1,2], Wouter Hoefsloot[2,10], Monique H. Reijers[2,10], Reinout van Crevel[1,2], Josephine S. van de Maat[1,2]*

1 Department of Internal Medicine, Radboud University Medical Centre, Nijmegen, The Netherlands, 2 Radboud Centre for Infectious Diseases (RCI), Radboud University Medical Centre, Nijmegen, The Netherlands, 3 Department of Research, Sint Maartenskliniek, Nijmegen, The Netherlands, 4 Department of Health Evidence, Section Biostatistics, Radboud University Medical Centre, Nijmegen, The Netherlands, 5 Centre of Expertise in Mycology, Radboud University Medical Centre, Canisius Wilhelmina Hospital, Nijmegen, The Netherlands, 6 Department of Internal Medicine, Bernhoven Hospital, Uden, The Netherlands, 7 Department of Internal Medicine, Canisius Wilhelmina Hospital, Nijmegen, The Netherlands, 8 Department of Internal Medicine, Jeroen Bosch Hospital, Den Bosch, The Netherlands, 9 Department of Internal Medicine, Sint Maartenskliniek, Nijmegen, The Netherlands, 10 Department of Pulmonary Diseases, Radboud University Medical Centre, Nijmegen, The Netherlands

* Josephine.vandemaat@radboudumc.nl

## Abstract

### Background

To date, survival data on risk factors for COVID-19 mortality in western Europe is limited, and none of the published survival studies have used a competing risk approach. This study aims to identify risk factors for in-hospital mortality in COVID-19 patients in the Netherlands, considering recovery as a competing risk.

### Methods

In this observational multicenter cohort study we included adults with PCR-confirmed SARS-CoV-2 infection that were admitted to one of five hospitals in the Netherlands (March to May 2020). We performed a competing risk survival analysis, presenting cause-specific hazard ratios ($HR_{CS}$) for the effect of preselected factors on the absolute risk of death and recovery.

### Results

1,006 patients were included (63.9% male; median age 69 years, IQR: 58–77). Patients were hospitalized for a median duration of 6 days (IQR: 3–13); 243 (24.6%) of them died, 689 (69.9%) recovered, and 74 (7.4%) were censored. Patients with higher age ($HR_{CS}$ 1.10, 95% CI 1.08–1.12), immunocompromised state ($HR_{CS}$ 1.46, 95% CI 1.08–1.98), who used anticoagulants or antiplatelet medication ($HR_{CS}$ 1.38, 95% CI 1.01–1.88), with higher

**Data Availability Statement:** All relevant data are within the manuscript and its Supporting Information files.

**Funding:** The author(s) received no specific funding for this work.

**Competing interests:** The authors have declared that no competing interests exist.

modified early warning score (MEWS) ($HR_{CS}$ 1.09, 95% CI 1.01–1.18), and higher blood LDH at time of admission ($HR_{CS}$ 6.68, 95% CI 1.95–22.8) had increased risk of death, whereas fever ($HR_{CS}$ 0.70, 95% CI 0.52–0.95) decreased risk of death. We found no increased mortality risk in male patients, high BMI or diabetes.

## Conclusion

Our competing risk survival analysis confirms specific risk factors for COVID-19 mortality in a the Netherlands, which can be used for prediction research, more intense in-hospital monitoring or prioritizing particular patients for new treatments or vaccination.

## Introduction

The pandemic of severe acute respiratory syndrome coronavirus 2 (SARS-CoV-2) associated coronavirus disease 2019 (COVID-19) causes significant morbidity and mortality worldwide. The first laboratory-confirmed case of COVID-19 in the Netherlands was reported on February 27, 2020 [1]. The disease initially spread across the southern provinces and rapidly disseminated further throughout the country. Currently, the Netherlands is amidst a second wave of infections and >350 thousand confirmed cases have been reported, including >7 thousand deaths [2, 3].

Studies describing the clinical features of COVID-19 and risk factors associated with incidence and timing of poor outcome have been extensively published [4], but survival data regarding risk factors for COVID-19 mortality in north-western Europe is limited. Furthermore, to our knowledge, none of the published survival studies have considered recovery as competing risk for mortality. Not taking competing risks into account leads to biased mortality estimates and to overestimation of survival curves [5, 6]. Analyzing mortality data with a more accurate competing risk analysis adds to the growing body of evidence on disease course and risk factors of a poor COVID-19 outcome. More robust knowledge about these risk factors is crucial to inform international prediction research [4]. In the current phase of the pandemic, now specific vaccines are underway, correct identification of patients at risk of severe disease and mortality is essential to prioritize patients for vaccination. In addition, while the virus still circulates in the population, it can guide clinical decisions on close monitoring, ICU admission or selection for new treatments. Therefore, this study aims to identify risk factors for in-hospital mortality in laboratory-confirmed COVID-19 patients in the Netherlands, using a competing risk approach.

## Materials and methods

In this observational, multicenter cohort study, we consecutively included adult patients with PCR-confirmed infection with SARS-CoV-2, who were admitted to one of five collaborating hospitals in the region of Gelderland and North-Brabant for at least 24 hours between March and May 2020. For the current analysis, we excluded patients if data regarding duration of hospital admission were missing for patients who died or recovered. A flow chart of inclusion is shown in S1 Fig. The study was approved by the institutional review board (IRB) of the Radboud university medical center (number 2020–2923 and 2020–6344). According to the IRB, only oral consent was required. Oral consent was obtained from all patients or their family and documented in the electronic medical records.

## Data collection and definitions

We extracted routine data from the electronic medical records, including baseline patient characteristics, information on clinical presentation, diagnostics, disease course, treatment and outcome. All patient data were entered anonymously into a web-based electronic case report form (using Castor Electronic Data Capture), only using a study identifier. The key linking patient information to study ID was saved in a local, protected file in the participating hospitals and not available to the researchers performing data analyses. Detailed methods on data collection and definitions are described in our parallel publication in this issue. For the current analysis, we used the Modified Early Warning Score (MEWS) as a summary measure of abnormal vital signs, based on body temperature, blood pressure, heart rate, respiratory rate, and the Alert Voice Pain Unresponsive (AVPU) score [7]. The AVPU score was not reported in the electronic medical records, so we used 'level of consciousness' as a proxy in the MEWS, with 'conscious' indicating Alert, and 'reduced level of consciousness' substituting Verbal, Pain and Unresponsive. We totaled duration of intensive care unit (ICU) admission for patients who were admitted to the ICU more than once. Readmissions within 4 weeks after initial discharge were considered extensions of the initial hospitalization. Follow-up duration was set to the duration until database lock for patients still admitted at the end of the study (n = 6). We defined death as in-hospital death or palliative discharge. Patients who were discharged due to clinical improvement, for medical rehabilitation, or transfer to a nursing home were considered 'recovered'. Patients who were transferred to a non-study hospital, who were lost to follow-up because of other reasons, or whose reason of discharge was unknown or missing were censored.

## Statistical analysis

Descriptive analyses were performed. Furthermore, we used a competing risks survival analysis approach to evaluate the effect of risk factors on the time from admission to death and on the time from admission to recovery [8, 9]. According to *Noordzij et al*, a competing risk occurs 'when subjects can experience one or more events or outcomes which 'compete' with the outcome of interest' [9]. In our study, recovery is considered a competing risk for mortality and is taken into account as an extra outcome, whereas in standard survival analyses, patients who recover are censored. However, the latter violates the assumption of noninformative censoring, i.e. the recovered patients are not representative of those who are still admitted to the hospital in terms of their risk of dying. Censoring recovered patients induces bias and overestimation of survival curves, i.e. Kaplan Meier will estimate incidence of death with upwards biases [8, 9]. For the competing risk analysis, we estimated univariable and multivariable cause-specific hazard ratios ($HR_{CS}$) for death and recovery for selected risk factors [5, 8]. These risk factors were pre-selected based on literature and expert opinion to be clinically relevant and routinely available at time of presentation, rather than based on statistical significance [10]. Furthermore, cumulative incidence probabilities were estimated using the Fine & Gray approach [5, 8]. Gray's test was used to compare equality of cumulative incidence curves (CIFs) across subgroups [11, 12]. The proportional hazards assumption was checked by an evaluation of the Schoenfeld residuals, as shown in S2 Fig.

Multivariable $HR_{CS}$ were also estimated for a subpopulation of patients with a 'non ICU admission policy'. The Netherlands executes a restrictive policy for ICU admission for patients who do not wish to be admitted to the ICU, or for whom the expected chance of recovery is thought to be too low or their overall prognosis too poor (e.g. because of comorbidity) to outweigh the potential harm of an ICU admission. The cause-specific hazard (CSH) model was fitted again for this subpopulation, to assess the influence of risk factors on the outcome in

these patients. Statistical analyses were conducted in R (v1.2.1578), using the 'survival', 'surv-miner' and 'cmprsk' packages for time-to-event analyses. Laboratory values were log-trans-formed to adjust for their skewed distribution.

### Missing data

Missing data in the variables 'non ICU admission policy, 'COVID treatment', 'oxygen supple-mentation' and 'ventilatory support' were considered as 'no' or 'none', assuming that it would be reported in the medical records if the concerning variable applied to the patient. Further-more, missing values in the variables 'X-ray', 'CT-scan', and 'blood culture' were considered as 'not performed', assuming that the results would be reported if patients had one of these tests. Missing values in all other variables were assumed to be missing at random (MAR) and were imputed using multiple imputation in ten datasets with the 'MICE' package. Analyses were performed on all ten imputed datasets and the results were pooled.

## Results

We included 1,006 PCR-confirmed COVID-19 patients (see flowchart of inclusion in S1 Fig). Their median age of 69 years (IQR: 58–77), most were male and presented with fever, cough and dyspnea, as shown in Table 1. Nearly all patients had one or more comorbidities. Of all patients, 243 (24.6%) died in-hospital or were discharged for palliative care, 689 (69.9%) recov-ered, and 74 (7.4%) were censored. The median duration of hospital admission until death and recovery was 6 days (IQR: 3–11) and 7 days (IQR: 4–13), respectively.

### Cumulative incidence curves

The CIF curves of the total population show that the probability of death after one, two and three weeks of hospital admission was 15.4% (95% 13.1–17.6), 20.5% (95% CI 18.0–23.0), and 22.4% (95% CI 19.8–25.0), respectively (Fig 1). The probabilities of recovery were 38.6% (95% CI 35.5–41.6), 54.1% (95% CI 51.0–57.2), and 60.3% (95% CI 57.3–63.3), respectively. Patients aged ≥70 years had a higher chance of death (p<0.001) and a lower chance of recovery (p<0.001) than patients aged <70 years (Fig 2). Males had a lower probability of recovery (p = 0.003) than females, but there was no statistical difference between both groups for death (p = 0.05) (Fig 3).

### Univariable and multivariable SH model

$HR_{CS}$ from univariable and multivariable SH models are reported in Table 2. Univariable analy-ses showed that older age increased the risk of death: with every year increase in age, the risk of death increased with 9% ($HR_{CS}$ 1.09, 95% CI 1.08–1.11), and the chances of recovery decreased with 1% ($HR_{CS}$ 0.99, 95% CI 0.98–0.99). Of the comorbidities, cardiovascular dis-ease, hypertension and pulmonary diseased increased risk of death. Patients with cardiovascu-lar disease had a 99% increase in the risk of death ($HR_{CS}$ 1.99, 95% CI 1.50–2.65), and a 17% decrease in the chances of recovery ($HR_{CS}$ 0.83, 95% CI 0.72–0.97). Patients with hypertension had a 39% increase in the risk of death ($HR_{CS}$ 1.39, 95% CI 1.07–1.79), and a 130% increase in chances of recovery ($HR_{CS}$ 2.30, 95% CI 1.98–2.68). Furthermore, patients with pulmonary disease had a 42% increase in risk of death ($HR_{CS}$ 1.42, 95% CI 1.08–1.88).

In terms of medication, both chronic use of anticoagulants or antiplatelet medication as well as ACE inhibitors or angiotensin-II receptor blockers were associated with an increased risk of death ($HR_{CS}$ 2.23, 95% CI 1.73–2.87, and $HR_{CS}$ 1.38, 95% CI 1.07–1.79, respectively). For symptoms, both fever and dyspnea decreased risk of death. Patients with fever had a 45%

**Table 1. Characteristics of the study population.**

| | Total (N = 1,006) *Median (IQR) or N(%)* | Missing *N(%)* |
|---|---|---|
| Age (years) | 69 (58–77) | 0 (0.0) |
| Sex, male | 643 (63.9) | 0 (0.0) |
| BMI | 27.6 (24.7–31.0) | 278 (27.6) |
| ≥1 comorbidity | 904 (90.1) | 3 (0.3) |
| Most frequent comorbidities [a] | | 3 (0.3) |
| Cardiovascular disease (incl. hypertension) | 582 (58.0) | |
| Hypertension | 390 (38.9) | |
| Pulmonary disease | 246 (24.5) | |
| Diabetes mellitus | 229 (22.8) | |
| Solid organ malignancies | 149 (14.9) | |
| Auto-immune disease | 122 (12.2) | |
| Chronic kidney disease | 112 (11.2) | |
| Immunocompromised [b] | 213 (21.3) | 4 (0.4) |
| Chronic use of ACE inhibitors and/or angiotensin-II receptor blockers | 336 (33.4) | 1 (0.1) |
| Chronic use of antiplatelet medication or anticoagulants | 374 (37.2) | 1 (0.1) |
| Non ICU admission policy | 312 (31.0) | 26 (2.6) |
| **Symptoms** | | |
| Duration of symptoms before admission (days) | 7 (5–10) | 85 (8.4) |
| Most frequent symptoms reported at admission [c] | | 8 (0.8) |
| Fever | 765 (76.7) | |
| Cough | 757 (75.9) | |
| Shortness of breath (dyspnea) | 696 (69.7) | |
| Fatigue | 354 (35.5) | |
| Diarrhea | 329 (33.0) | |
| **Vital signs and physical examination at time of admission** | | |
| Ill appearance | 399 (56.7) | 302 (30.0) |
| Dyspnea | 336 (55.5) | 401 (39.9) |
| Fever [d] | 459 (47.2) | 33 (3.3) |
| Blood pressure (mmHg) [e] | | 32 (3.2) |
| Normal | 862 (88.5) | |
| Hypotension | 7 (0.7) | |
| Hypertension | 105 (10.8) | |
| Tachycardia [f] | 278 (28.5) | 30 (3.0) |
| Tachypnea [g] | 573 (60.1) | 53 (5.3) |
| Hypoxia [h] | 149 (15.1) | 22 (2.2) |
| Modified Early Warning Score (MEWS) [i] | 3 (2–4) | 76 (7.6) |
| **Laboratory parameters at time of admission** | | |
| Hemoglobin (mmol/L) | 8.6 (7.8–9.2) | 32 (3.2) |
| White blood cell count (*$10^9$/L) | 6.7 (5.0–9.0) | 31 (3.1) |
| Neutrophil count (*$10^9$/L) | 5.2 (3.5–7.3) | 87 (8.6) |
| Lymphocyte count (*$10^9$/L) | 0.9 (0.6–1.2) | 84 (8.3) |
| Thrombocytes (*$10^9$/L) | 205 (159–268) | 40 (4.0) |
| C-Reactive Protein (mg/L) | 92.0 (47.0–147.0) | 28 (2.8) |
| Ferritin (μg/L) | 796.5 (408.8–1466.2) | 206 (20.5) |
| D-dimer (ng/L) [j] | 920.0 (500.0–1810.0) | 194 (19.3) |
| Lactate dehydrogenase (U/L) | 359.0 (278.5–466.5) | 103 (10.2) |
| Procalcitonin (μg/L) | 0.15 (0.08–0.33) | 412 (41.0) |
| Creatinine (μmol/L) | 84.0 (68.0–106.0) | 34 (3.4) |
| **Diagnostics** | | |
| Result first PCR, positive | 924 (93.8) | 21 (2.1) |

*(Continued)*

**Table 1.** (Continued)

| | Total (N = 1,006) *Median (IQR) or N(%)* | Missing *N(%)* |
|---|---|---|
| Second PCR performed | 218 (21.7) | 12 (1.2) |
| *Of which: result second PCR positive* | *133 (61.0)* | *4 (1.8)* |
| Chest X-Ray performed | 608 (61.6) | 19 (1.9) |
| *Of which: chest X-ray suggestive for COVID-19* | *325 (53.5)* | *208 (34.2)* |
| CT scan performed | 500 (50.9) | 23 (2.3) |
| *Of which: CORADS classification* | *44 (11.5)* | *125 (25.0)* |
| *1–3 (not suggestive for COVID-19)* | *337 (88.5)* | *158 (31.6)* |
| *4–5 (suggestive for COVID-19)* | *12 (9–15)* | |
| *Of which: CT severity score* | | |
| Discharge | | |
| Duration of hospital admission (days) | 6 (3–13) | 6 (0.6) |
| Reason for discharge | | 20 (2.0) |
| Clinical improvement | 689 (69.9) | |
| Patient deceased in-hospital | 238 (24.1) | |
| Palliative discharge | 5 (0.5) | |
| Transfer to a non-study hospital | 44 (4.5) | |
| Other | 8 (0.8) | |
| Unknown | 2 (0.2) | |
| Readmission after initial discharge | 48 (4.8) | 0 (0.0) |
| ICU admission | 207 (21.2) | 30 (3.0) |
| Duration of hospital admission to first ICU admission (days) | 2.0 (0.5–4.0) | 0 (0.0) |
| Duration of ICU admission (days) | 17 (10–27) | 13 (6.3) |
| Invasive ventilatory support (incl. prone position) | 184 (18.4) | 5 (0.5) |
| Complications [k] | 640 (63.6) | 0 (0.0) |

Abbreviations: N = number, IQR = interquartile range, BMI = body mass index

[a] Comorbidities present in >10% of the population

[b] Immunocompromised was defined as having a haematologic malignancy, stem cell or organ transplantation, auto-immune disease, HIV/AIDS and/or use of immunosuppressive medication.

[c] five most commonly reported symptoms

[d] Fever was defined as a body temperature $\geq$ 38.0˚C.

[e] Hypertension was defined as: systolic blood pressure $\geq$140 mmHg and diastolic blood pressure $\geq$90 mmHg.

Hypotension was defined as: systolic blood pressure <90 mmHg and diastolic blood pressure <60 mmHg.

[f] Tachycardia was defined as a heart rate >100 bpm.

[g] Tachypnea was defined as a respiratory rate >20 bpm.

[h] Hypoxia was defined as a peripheral oxygen saturation <90%.

[i] Adjusted version of Modified Early Warning Score.

[j] D-dimer was tested within 24h of hospital admission.

[k] The most common complications included acute kidney injury (18.2%), delirium (13.0%), and new onset/episode of atrial fibrillation (9.6%).

decrease in risk of death (HR$_{CS}$ 0.55, 95% CI 0.42–0.72) and patients with dyspnea a 25% decrease in risk of death (HR$_{CS}$ 0.75, 95% CI 0.57–0.98). In terms of diagnostics, neutrophil-to-lymphocyte rate, creatinine levels and ferritin levels at time of admission were associated with risk of death and risk of recovery. A ten-fold increase in the neutrophil-to-lymphocyte rate was associated with a 58% increase in risk of death (HR$_{CS}$ 1.58, 95% CI 1.02–2.42) and a 35% decrease in the chance of recovery (HR$_{CS}$ 0.65, 95% CI 0.50–0.85). Furthermore, a ten-fold increase in creatinine levels at time of admission were associated with a 444% increased risk of death (HR$_{CS}$ 5.44, 95% CI 3.13–9.44) and a 37% decrease in chances of recovery (HR$_{CS}$ 0.63, 95% CI 0.40–0.99). Finally, a ten-fold increase in ferritin levels at time of admission was associated with a 31% decrease in risk of death (HR$_{CS}$ 0.69, 95% CI 0.50–0.96), and a 42% decrease in chances of recovery (HR$_{CS}$ 0.58, 95% CI 0.49–0.70).

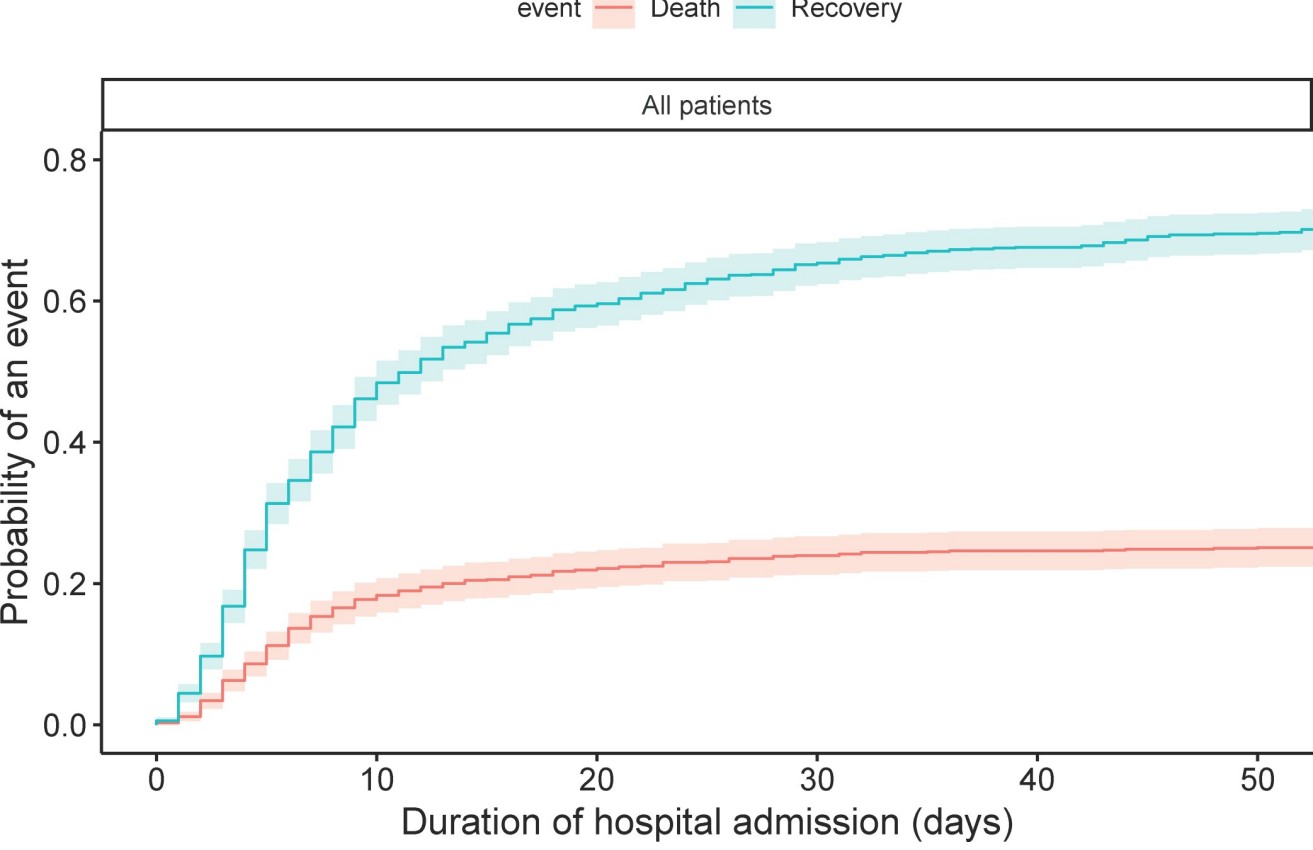

**Fig 1. Cumulative incidence plot of death and recovery in the total population.** The probability of death conditional on not having recovered after one, two and three weeks of hospital admission was 15.4% (95% 13.1–17.6), 20.5% (95% CI 18.0–23.0), and 22.4% (95% CI 19.8–25.0), respectively. The probability of recovery conditional on not having died after one, two and three weeks of hospital admission was 38.6% (95% CI 35.5–41.6), 54.1% (95% CI 51.0–57.2), and 60.3% (95% CI 57.3–63.3).

In multivariable analyses, the following factors were associated with risk of death: age, immunocompromised state, chronic use of anticoagulants or antiplatelet medication, fever, MEWS, lactate dehydrogenase values at time of admission, and ferritin values at time of admission. Firstly, with every year increase in age, the risk of death increased with 10% (HR$_{CS}$ 1.10, 95% CI 1.08–1.12), and the chances of recovery decreased with 1% (HR$_{CS}$ 0.99, 95% CI 0.98–0.99). In other words, if in two patients all variables except for age are the same, the patient who is one year older has a 10% higher risk of dying. Furthermore, patients with immunocompromised state had a 46% increased risk of death (HR$_{CS}$ 1.46, 95% CI 1.08–1.98), and 24% decrease in chances of recovery (HR$_{CS}$ 0.76, 95% CI 0.62–0.93). Moreover, patients that used anticoagulants or antiplatelet medication had a 38% increase in risk of death (HR$_{CS}$ 1.38, 95% CI 1.01–1.88). Furthermore, patients with fever had a 30% decrease in risk of death (HR$_{CS}$ 0.70, 95% CI 0.52–0.95). In terms of diagnostics, with every unit increase of MEWS, the risk of death increased with 9% (HR$_{CS}$ 1.09, 95% CI 1.01–1.18). Moreover, a ten-fold increase in lactate dehydrogenase values at time of admission was associated with a 568% increase in risk of death (HR$_{CS}$ 6.68, 95% CI 1.95–22.8), and a 75% decrease in chances of recovery (HR$_{CS}$ 0.25, 95% CI 0.13–0.48). Finally, a ten-fold increase in ferritin values at time of admission was associated with a 23% decrease in chances of recovery (HR$_{CS}$ 0.77, 95% CI 0.60–0.99).

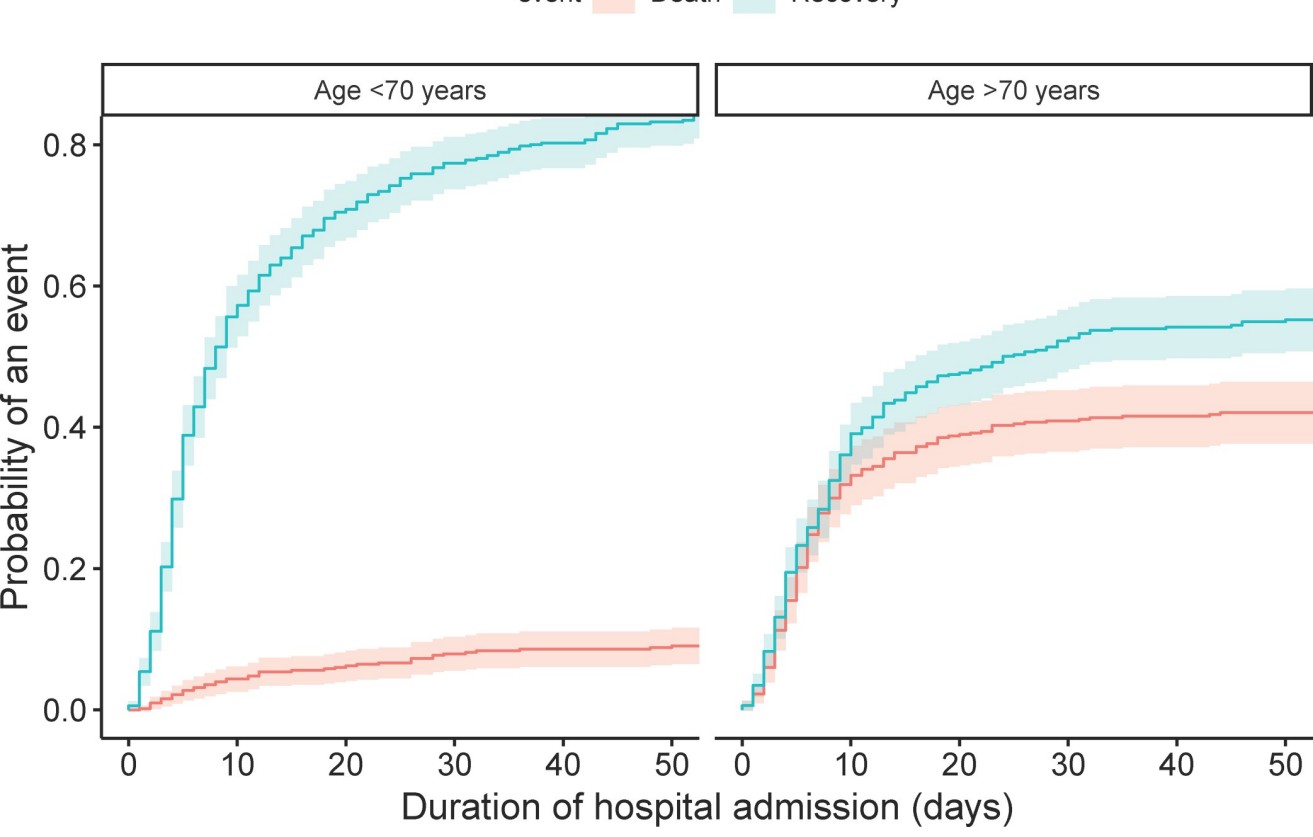

**Fig 2. Cumulative incidence plot of death and recovery in the total population, separated by age group.** Gray's test indicated a significant difference between two groups for both death (p<0.001) and recovery (p<0.001). The probability of death for patients aged <70 years after one, two and three weeks of hospital admission was 3.6% (95% CI 2–5.2), 5.4% (95% CI 3.4–7.4), and 6.5% (95% CI 4.3–8.6), respectively, whereas for patients aged ≥70 years, the probability of death was 27.8% (95% CI 23.8–31.8), 36.4% (95% CI 32.1–40.7), and 39.2% (95% CI 34.8–43.5), respectively.

### Non ICU admission policy

In this study, 318 (31.5%) patients had a non ICU admission policy, of whom 199 (62.6%) were male, with a median age of 79 years (IQR 74–83), and a median MEWS of 2 (IQR 2–4). Of these patients, 151 (47.5%) recovered, 158 (49.7%) died and 9 (2.8%) were censored. Multivariable analyses revealed no statistically significant associations with risk of death, but that patients with longer symptom duration had an increased chance of recovery (HR$_{CS}$ 1.02, 95% CI 1.00–1.04) and patients with symptoms of dyspnea had a decreased chance of recovery (HR$_{CS}$ 0.65, 95% CI 0.44–0.96).

## Discussion

### Summary of findings

In this Dutch hospital population, approximately a quarter of all COVID-19 patients died after a median hospital admission of six days during the first wave of the epidemic. Using a competing risk approach, we identified age, comorbidities, such as cardiovascular diseases, symptoms, and abnormal laboratory values as the most important risk factors in univariate analyses. After adjusting for all relevant factors at baseline, we found that higher age, immunocompromised state, chronic use of anticoagulants or antiplatelet medication, higher MEWS, and higher

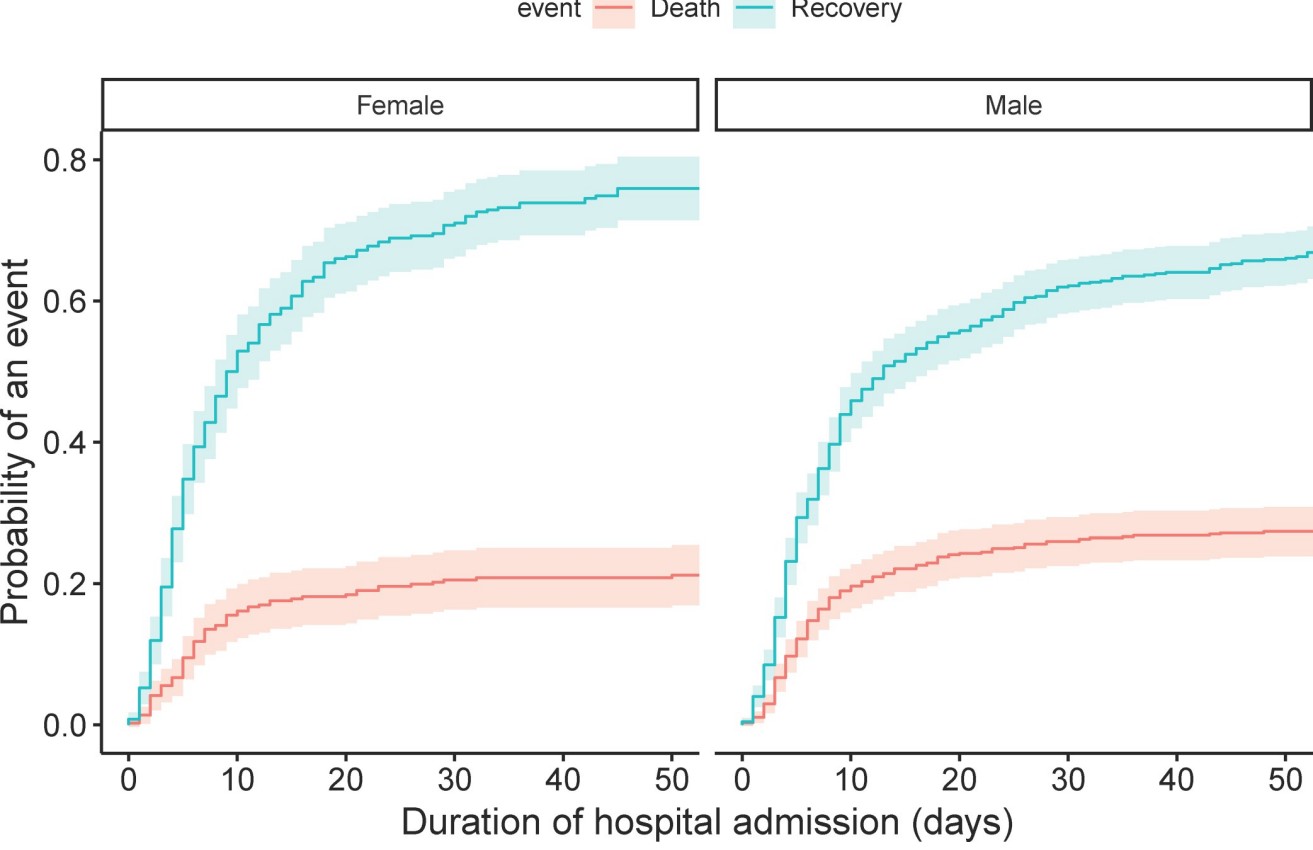

**Fig 3. Cumulative incidence plot of death and recovery in the total population, separated by sex.** Gray's test indicated a statistically significant difference between both groups for recovery (p = 0.003), but not for death (p = 0.050). The probability of death for females after one, two and three weeks of hospital admission was 12.5% (95% CI 10.0–17.1), 17.6% (95% CI 13.7–21.6), and 19.1% (95% CI 15.0–23.2), respectively, whereas for males the probability of death was 16.4% (95% CI 13.5–19.3), 22.1% (95% CI 18.9–25.4), and 24.3% (95% CI 20.9–27.6), respectively.

values of LDH at time of hospital admission were associated with increased risk of death. On the other hand, fever at time of admission was associated with lower mortality. Male sex was not a significant risk factor for mortality.

## Interpretation of results and comparison to literature

Conventional survival analyses do not take competing risks into account, which leads to biased mortality estimates and to overestimation of survival curves. Using the competing risk approach, we took into account that patients who recovered were no longer at the same risk of dying than those who remained hospitalized, resulting in less biased mortality estimates. Even though death or recovery are the two possible final outcomes of the disease, the time to death and time to recovery may not the same (i.e. time to death was generally shorter than time to recovery). In addition, 10% of our patients were censored due to transfer to other hospitals. As a result, the risk factors influencing death may differ from risk factors for recovery [13]. For example, we identified strong risk factors that both increased the risk of dying and reduced the recovery risk, namely higher age, immunosuppression and high LDH values at admission. At the same time, use of anticoagulant or antiplatelet medication and a high MEWS were only associated with an increased risk of dying. Fever at admission reduced this risk, and ferritin showed equivocal results, reducing the risk of death as well as the risk of recovery.

**Table 2. Univariable and multivariable cause-specific hazard ratios (HR_CS) including 95% confidence intervals for death and recovery.**

| | Univariable | | Multivariable | |
|---|---|---|---|---|
| | **Death** | **Recovery** | **Death** | **Recovery** |
| Age (years) | 1.09 (1.08–1.11) * | 0.99 (0.98–0.99) * | 1.10 (1.08–1.12) * | 0.99 (0.98–0.99) * |
| Sex, male | 1.15 (0.88–1.52) | 0.80 (0.68–0.93) * | 1.07 (0.79–1.47) | 0.90 (0.75–1.08) |
| BMI | 0.99 (0.96–1.01) | 1.01 (0.99–1.03) | 1.01 (0.98–1.04) | 1.01 (0.99–1.03) |
| Diabetes Mellitus | 1.23 (0.93–1.63) | 0.89 (0.74–1.07) | 1.17 (0.86–1.59) | 0.85 (0.69–1.04) |
| Cardiovascular disease (incl. hypertension) | 1.99 (1.50–2.65) * | 0.83 (0.72–0.97) * | 1.05 (0.69–1.59) | 0.71 (0.54–0.93) * |
| Hypertension | 1.39 (1.07–1.79) * | 2.30 (1.98–2.68) * | 0.78 (0.56–1.10) | 1.15 (0.90–1.48) |
| Pulmonary disease | 1.42 (1.08–1.88) * | 1.01 (0.85–1.21) | 1.33 (0.98–1.80) | 0.88 (0.73–1.07) |
| Immunocompromised [a] | 1.31 (0.99–1.74) | 0.87 (0.72–1.05) | 1.46 (1.08–1.98) * | 0.76 (0.62–0.93) * |
| Chronic use of anticoagulant or antiplatelet medication | 2.23 (1.73–2.87) * | 1.01 (0.86–1.18) | 1.38 (1.01–1.88) * | 1.15 (0.93–1.43) |
| Chronic use of ACE inhibitors and/or angiotensin II receptor blockers | 1.38 (1.07–1.79) * | 0.97 (0.83–1.14) | 0.99 (0.74–1.32) | 1.09 (0.88–1.33) |
| Chest X-Ray | | | | |
| Performed, not suggestive for COVID-19 | Ref | Ref | Ref | Ref |
| Performed, suggestive for COVID-19 | 0.96 (0.59–1.55) | 0.96 (0.71–1.28) | 1.07 (0.63–1.80) | 1.52 (1.05–2.19) * |
| Not performed | 0.76 (0.48–1.20) | 1.02 (0.75–1.39) | 0.90 (0.54–1.48) | 1.18 (0.82–1.70) |
| CT scan severity score | 0.99 (0.97–1.02) | 0.95 (0.93–0.97) * | 1.01 (0.98–1.05) | 0.97 (0.95–0.99) * |
| Symptom duration (days) | 0.98 (0.95–1.00) | 1.01 (1.00–1.02) | 0.98 (0.96–1.01) | 1.01 (1.00–1.02) * |
| Symptoms, fever | 0.55 (0.42–0.72) * | 0.95 (0.79–1.15) | 0.70 (0.52–0.95) * | 1.05 (0.85–1.30) |
| Symptoms, dyspnea | 0.75 (0.57–0.98) * | 0.94 (0.80–1.11) | 0.77 (0.58–1.03) | 1.01 (0.84–1.21) |
| Modified Early Warning Score (MEWS) [b] | 1.06 (0.99–1.14) | 0.95 (0.91–1.00) * | 1.09 (1.01–1.18) * | 0.97 (0.93–1.02) |
| Neutrophil-to-lymphocyte rate [c] | 1.58 (1.02–2.42) * | 0.65 (0.50–0.85) * | 0.97 (0.59–1.60) | 1.01 (0.74–1.38) |
| Lactate dehydrogenase (U/L) [c] | 1.67 (0.76–3.65) | 0.16 (0.10–0.25) * | 6.68 (1.95–22.8) * | 0.25 (0.13–0.48) * |
| Creatinine (μmol/L) [c] | 5.44 (3.13–9.44) * | 0.63 (0.40–0.99) * | 1.84 (0.87–3.90) | 1.17 (0.69–1.99) |
| Procalcitonin (μg/L) [c] | 1.27 (0.98–1.64) | 0.78 (0.68–0.90) * | 1.04 (0.76–1.41) | 0.88 (0.75–1.03) |
| C-reactive protein (mg/L) [c] | 1.16 (0.82–1.63) | 0.61 (0.51–0.74) * | 1.25 (0.79–2.00) | 0.88 (0.69–1.11) |
| Ferritin (μg/L) [c] | 0.69 (0.50–0.96) * | 0.58 (0.49–0.70) * | 0.66 (0.43–1.02) | 0.77 (0.60–0.99) * |
| D-dimer (ng/L) [d] | 1.10 (0.94–1.30) | 0.88 (0.83–0.93) * | 0.99 (0.84–1.16) | 0.94 (0.87–1.00) |

* Statistically significant, i.e. p<0.05.

[a] Immunocompromised was defined as having an hematologic malignancy, stem cell or organ transplantation, auto-immune disease, HIV/AIDS and/or use of immunosuppressive medication.

[b] Adjusted version of Modified Early Warning Score.

[c] These laboratory variables are log-transformed. HRs should be interpreted for a 10-fold increase in the concerning variable, rather than a one unit increase.

[d] D-dimer is determined within 24h of hospital admission.

In our study population, approximately 25% of all patients died, which is consistent with other studies [14, 15]. Our study showed that age and higher MEWS at time of admission were risk factors for in-hospital mortality, which is in line with results of many other studies [4, 16–18]. Higher blood LDH levels at time of admission increased the probability of death and decreased the chance of recovery, which is in line with a growing body of evidence [4, 19, 20]. We included both cardiovascular disease and hypertension in our model, which may be subject to collinearity. However, a sensitivity analyses excluding hypertension from the model did not change our estimates and standard errors, showing that our reported estimates are valid. We found immunosuppression as a risk factor for mortality. This has been reported previously, although the exact role of the immune system in COVID-19 is complex [16, 21, 22]. Poor outcome could be determined by a declining immune system less able to clear the virus, but lung tissue damage in severe cases could also be caused by an exaggerated immune

response, rather than damage inflicted by the virus itself [23–26]. Consequently, immunocompromised patients may be protected from this type of hyperinflammation [21, 27]. We found that chronic use of anticoagulants or antiplatelet medication was associated with increased risk of death. Although this may be partially explained by the fact that these medications are used by patients with cardiovascular disease, which has been reported as an individual risk factor for in-hospital mortality [4], anticoagulant and/or antiplatelet medication remained an independent risk factor for death in our multivariable analyses. Thromboembolic events are frequently reported in association with severe COVID-19 disease and mortality [28, 29], and current guidelines suggest prophylactic anticoagulants in all hospitalized COVID-19 patients if not contraindicated. However, studies have reported conflicting results regarding the effect of anticoagulants/antiplatelet medication on COVID-19 mortality [30], ranging from a protective effect [31] to a harmful effect [32, 33], or no association [31, 34]. Prospective studies and RCTs are needed to explore the true effects of these medications in hospitalized COVID-19 patients.

Findings of a living systematic review of 23 prognostic studies about COVID-19 mortality indicated that age, immunocompromising comorbidities, composite scores of vital parameters and blood LDH are frequently reported predictors of in-hospital mortality, similar to our findings [4]. Blood ferritin levels and anticoagulants or antiplatelet medication were scarcely reported [4]. On the other hand, male sex and comorbidities, such as cardiovascular and pulmonary diseases, that were significantly associated with in-hospital mortality in our univariable but not multivariable analyses, were frequently reported in other prognostic studies. This is an important finding and shows that many of these risk factors are interacting with each other. Last, we found no increased risk of dying in patients with male sex, high BMI or diabetes, which are also risk factors that have been reported previously. Larger studies may be needed to reveal additional risk factors of clinical importance.

## Strengths and limitations

A strength of this study is the large, multicenter population in an area in which infections were clustered. We analyzed mortality in COVID-19 patients using a competing risk approach, leading to more accurate risk estimates for mortality than when using conventional survival analysis. As previously explained, conventional survival analyses may have resulted in biased estimates and Kaplan-Meier curves presenting overestimated incidence of death. There were some limitations to this study. First, approximately 10% of patients in our study were censored, mainly due to frequent transfer of patients. However, this was considered within acceptable limits [35–37]. Second, our data consisted of routinely collected data, resulting in missing values in several variables. We used multiple imputation to minimize associated bias. The advantage of routinely collected data is that the study's predictors are readily available and be used in clinical practice without extra effort. All reported predictors are variables measured at time of hospital admission, which means that these factors can be used to identify patients in need of more intensive care and monitoring in an early stage of hospitalization. Finally, data were collected in multiple centers with differences in manner of reporting in medical records and differences in clinical management. The advantage of this multicenter approach is that it increased the generalizability of our findings.

## Conclusion

Using a robust, competing risk survival analysis, our study confirmed specific risk factors for in-hospital COVID-19 mortality, adding rigor to the current knowledge of risk factors. We confirmed that age, immunocompromised state, use of anticoagulants or antiplatelet

medication, MEWS, blood LDH are important risk factors for death in hospitalized COVID-19 patients. The risk factors we identified are mostly in line with the growing body of evidence on COVID-19, although we did not find evidence for increased mortality risk in male patients or those with high BMI or diabetes. All identified risk factors are routinely available at time of admission. They can be used to guide clinical decisions on intense monitoring of patients in-hospital, or to select patients that may benefit most from new treatments. In addition, the more general risk factors like age and comorbidities could be used to prioritize patients for vaccinations in current times where vaccines are still scarce. Lastly, our risk factor analysis provides further input for international prediction research for mortality of COVID-19.

## Supporting information

**S1 Checklist. STROBE statement—checklist of items that should be included in reports of *cohort studies*.**
(PDF)

**S1 Fig. Flowchart of inclusion.**
(PDF)

**S2 Fig. Schoenfeld residuals plots.** A. Plots of the Schoenfeld residuals for the multivariable CSH model for death. B. Plots of the Schoenfeld residuals for the multivariable CSH model for recovery.
(PDF)

**S1 Dataset. Anonymized dataset.**
(CSV)

## Acknowledgments

We thank the RCI-COVID-19 study group for clinical input and critical feedback. Furthermore, we thank all students and nurses that helped to extract data from electronic records.

## Author Contributions

**Conceptualization:** Gerine Nijman, Maike Wientjes, Jordache Ramjith, Jacobien Hoogerwerf, Reinout van Crevel, Josephine S. van de Maat.

**Data curation:** Gerine Nijman, Nico Janssen, Evertine Abbink, Marc Blaauw, Ton Dofferhoff, Marjan van Apeldoorn, Karin Veerman, Josephine S. van de Maat.

**Formal analysis:** Gerine Nijman, Maike Wientjes, Josephine S. van de Maat.

**Methodology:** Gerine Nijman, Jordache Ramjith, Josephine S. van de Maat.

**Project administration:** Evertine Abbink.

**Supervision:** Maike Wientjes, Jacobien Hoogerwerf, Quirijn de Mast, Jaap ten Oever, Monique H. Reijers, Reinout van Crevel, Josephine S. van de Maat.

**Writing – original draft:** Gerine Nijman.

**Writing – review & editing:** Maike Wientjes, Jordache Ramjith, Nico Janssen, Jacobien Hoogerwerf, Evertine Abbink, Marc Blaauw, Ton Dofferhoff, Marjan van Apeldoorn, Karin Veerman, Quirijn de Mast, Jaap ten Oever, Wouter Hoefsloot, Monique H. Reijers, Reinout van Crevel, Josephine S. van de Maat.

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
