## [Decision Letter · Decision Letter 0]

17 Feb 2021

PONE-D-20-40238

Risk factors for in-hospital mortality in laboratory-confirmed COVID-19 patients in the Netherlands: a competing risk survival analysis.

PLOS ONE

Dear Dr. Josephine S. van de Maat

Thank you for submitting your manuscript to PLOS ONE. After careful consideration, we feel that it has merit but does not fully meet PLOS ONE’s publication criteria as it currently stands. Therefore, we invite you to submit a revised version of the manuscript that addresses the points raised during the review process.

We look forward to receiving your revised manuscript.

Kind regards,

Francesco Di Gennaro

Academic Editor

PLOS ONE

Journal Requirements:

2. Thank you for including your ethics statement:  "The study was reviewed by the institutional review board of the Radboud university medical center (number 2020-2923 and 2020-6344). Verbal informed consent was obtained from all patients or their family. ".   

Please amend your current ethics statement to confirm that your named institutional review board or ethics committee specifically approved this study.

3. In the Methods, please provide further clarifications on the following:

- Why written consent could not be obtained

- Whether the Institutional Review Board (IRB) approved use of oral consent

- How oral consent was documented

For more information, please see our guidelines for human subjects research: https://journals.plos.org/plosone/s/submission-guidelines#loc-human-subjects-research.

4. In ethics statement in the manuscript and in the online submission form, please provide additional information about the patient records/samples used in your retrospective study. Specifically, please ensure that you have discussed whether all data/samples were fully anonymized before you accessed them.

Additional Editor Comments:

dear authors follow reviewer suggestion to improve your paper

Reviewers' comments:

Reviewer's Responses to Questions

**Comments to the Author**

1. Is the manuscript technically sound, and do the data support the conclusions?

Reviewer #1: Yes

Reviewer #2: Yes

2. Has the statistical analysis been performed appropriately and rigorously? 

Reviewer #1: Yes

Reviewer #2: Yes

3. Have the authors made all data underlying the findings in their manuscript fully available?

Reviewer #1: Yes

Reviewer #2: Yes

4. Is the manuscript presented in an intelligible fashion and written in standard English?

Reviewer #1: Yes

Reviewer #2: Yes

5. Review Comments to the Author

Reviewer #1: the data about COVID-19 risk factors and prognostic indications are far from being satisfactory. Any study that help to understand factors that influence prognosis is welcomed. The current study discusses many of these factors. It is now expected to cover all of them andeven each factor can have several sub-risk factors. For example if you discuss diabetes, type of diabetes, duration, degree of control, comorbidies and complications can be risk factors to be discussed. Therefore, each reviewer may have additonal factors to be discussed but what has been analysed in this study is comperhensive enough. However for the application of the data you mentioned in the aim that the data can be used to define priorities for vaccination and intensive treatment. I recommed you subdivide the risk factors into two groups, first the risk factors to define priorities for vaccination like age, BMI, diabetes, autoimmune disease, etc. this will help decision makers define priorities for vaccination which can be critical, especially in countries with limitted resources where mass vccinations is not feasible in the short term. Second is the risk factors for already infected subjects like duration of cough fever diarrhea, PCR, lymphocyte count, etc. Such a data will make the study useful for practical application and may be a nucleus with which other similar studies can establish a score system for risk of serious infection and risk of mortality among infected subjects.

Reviewer #2: In this well performed and interesting study, Authors used a competing risk survival analysis to evaluate predictors of death in a large cohort of Dutch patients.

Overall, the analysis is sound, robust, and well described.

A few minor concerns are listed below:

1) In multivariable model (Table 2) “cardiovascular diseases (incl. hypertension)” and “hypertension” were both included in the model; collinearity has been investigated in this case?

2) “Use of anticoagulants or antiplatelet medication” resulted connected with an increased risk of death. According to Authors hypothesis this association could be explained by the underlying cardiovascular disease.

However: i) in multivariable model cardiovascular disease lost the statistically significance, implying that the use of anticoagulants/antiplatelets was independently associated with mortality;

ii) a considerable risk of thromboembolic events was reported in course of COVID-19 (Bavaro DF, et al. Occurrence of Acute Pulmonary Embolism in COVID-19-A case series. Int J Infect Dis. 2020;98:225-226.).

In my opinion, this work (or similar ones) should be cited, and the risk of death according to use of anticoagulants should be better discussed since current guidelines suggest the use of low-molecular-weight-heparin in all hospitalized COVID-19 patients, if not contraindicated.

6. PLOS authors have the option to publish the peer review history of their article (what does this mean?). If published, this will include your full peer review and any attached files.

Reviewer #1: **Yes: **Aly Ahmed Abdel Rahim

Reviewer #2: No

---

## [Author Response · Author response to Decision Letter 0]

3 Mar 2021

Ref: PONE-D-20-40238

Title: Risk factors for in-hospital mortality in laboratory-confirmed COVID-19 patients in the Netherlands: a competing risk survival analysis

Nijmegen, the Netherlands

25 February 2021

Dear Francesco Di Gennaro, academic editor of PLoS ONE,

We are pleased to hear that our manuscript entitled “Risk factors for in-hospital mortality in laboratory-confirmed COVID-19 patients in the Netherlands: a competing risk survival analysis“ is considered for publication in PLoS ONE. The reviewers acknowledged the importance of our study, and mentioned that our analyses were comprehensive, robust and well described. We thank the reviewers for their valuable comments to improve the paper, we have taken the opportunity to clarify some issues raised, and have revised our manuscript according to their suggestions. 

Our response to the individual review items can be found below. 

We hope our manuscript is now suitable for publication in PLoS ONE. 

On behalf of the co-authors,

Yours sincerely,

Gerine Nijman and Josephine van de Maat

Comment #1: Please ensure that your manuscript meets PLOS ONE's style requirements, including those for file naming. 

Response #1: The main body and author affiliations now meet PLoS ONE’s style requirements, including those for file naming. 

Comment #2: Thank you for including your ethics statement: "The study was reviewed by the institutional review board of the Radboud university medical center (number 2020-2923 and 2020-6344). Verbal informed consent was obtained from all patients or their family. ". Please amend your current ethics statement to confirm that your named institutional review board or ethics committee specifically approved this study. Once you have amended this/these statement(s) in the Methods section of the manuscript, please add the same text to the “Ethics Statement” field of the submission form (via “Edit Submission”). 

Response #2: We have adjusted the ethics statement in the methods section to clarify this and have amended the current ethics statement, Methods section, line 91-92: 

“The study was approved by the institutional review board (IRB) of the Radboud university medical center (number 2020-2923 and 2020-6344).”

Comment #3: In the Methods, please provide further clarifications on the following:

- Why written consent could not be obtained

- Whether the Institutional Review Board (IRB) approved use of oral consent

- How oral consent was documented

Response #3: The institutional review board (IRB) of the Radboudumc waived the need for written consent and therefore approved the use of oral consent for this study. Oral consent was obtained from all patients or their family and documented in the electronical medical records. We have changed the methods section to clarify this, line 93-95: 

“According to the IRB, only oral consent was required. Oral consent was obtained from all patients or their family and documented in the electronic medical records.”.

Comment #4: In ethics statement in the manuscript and in the online submission form, please provide additional information about the patient records/samples used in your retrospective study. Specifically, please ensure that you have discussed whether all data/samples were fully anonymized before you accessed them.

Response #4: All patient data were entered anonymously into the electronic case report forms (eCRF; Castor Electronic Data Capture). The authors that were involved in the data analysis only received coded data without traceable information leading to patients involved in this study. 

We have clarified this in the methods section, line 101-102: 

“All patient data were entered anonymously into a web-based electronic case report form (using Castor Electronic Data Capture), only using a study identifier. The key linking patient information to study ID was saved in a local, protected file in the participating hospitals and not available to the researchers performing data analyses.”

Comment #5: We note that you have indicated that data from this study are available upon request. PLOS only allows data to be available upon request if there are legal or ethical restrictions on sharing data publicly. For information on unacceptable data access restrictions, please see http://journals.plos.org/plosone/s/data-availability#loc-unacceptable-data-access-restrictions. In your revised cover letter, please address the following prompts:

b) If there are no restrictions, please upload the minimal anonymized data set necessary to replicate your study findings as either Supporting Information files or to a stable, public repository and provide us with the relevant URLs, DOIs, or accession numbers. 

Response #5: In our re-submission we have uploaded a supporting information file with the minimal required anonymized data to reproduce our study findings (S1 Dataset). 

Reviewer #1:

The data about COVID-19 risk factors and prognostic indications are far from being satisfactory. Any study that help to understand factors that influence prognosis is welcomed. The current study discusses many of these factors. It is now expected to cover all of them and even each factor can have several sub-risk factors. For example if you discuss diabetes, type of diabetes, duration, degree of control, comorbidies and complications can be risk factors to be discussed. Therefore, each reviewer may have additonal factors to be discussed but what has been analysed in this study is comprehensive enough. However for the application of the data you mentioned in the aim that the data can be used to define priorities for vaccination and intensive treatment. I recommend you subdivide the risk factors into two groups, first the risk factors to define priorities for vaccination like age, BMI, diabetes, autoimmune disease, etc. this will help decision makers define priorities for vaccination which can be critical, especially in countries with limited resources where mass vaccinations is not feasible in the short term. Second is the risk factors for already infected subjects like duration of cough fever diarrhea, PCR, lymphocyte count, etc. Such a data will make the study useful for practical application and may be a nucleus with which other similar studies can establish a score system for risk of serious infection and risk of mortality among infected subjects.

Response: We thank the reviewer for carefully examining our manuscript. The reviewer recommends to subdivide risk factors into two groups; (1) risk factors to define priorities for vaccination, i.e. in patients without a confirmed COVID-19 infection; and (2) risk factors to define priorities for intensive treatment, i.e. in patients who are already infected. 

In our study population, this subdivision is not possible, as we included a population that solely consisted of laboratory-confirmed COVID-19 patients (non-infected patients were not included). Therefore, we could not make a separate survival model to describe risk factors for patients that were not yet infected at time of inclusion. Nevertheless, we still think our results relevant to define prioritized target groups for both intensive treatment and vaccination. COVID-19 vaccination is, among other things, aimed at protection against severe disease and reducing mortality. This study helps to identify groups of patients who are at most at risk of death (once hospitalized), and in whom prevention of the disease is most urgent. Consequently, these groups of patients may be prioritized as target groups for vaccination. We agree with the reviewer that certain risk factors cannot be used for this purpose (e.g. abnormal lab values), so these factors only apply to the already hospitalized population in whom decisions need to be made on intensive monitoring and treatment. 

We added this consideration to the discussion section of our revised manuscript, line 379 – 380: 

“In addition, the more general risk factors like age and comorbidities could be used to prioritize patients for vaccinations in current times where vaccines are still scarce.”

Reviewer #2:

In this well performed and interesting study, authors used a competing risk survival analysis to evaluate predictors of death in a large cohort of Dutch patients. Overall, the analysis is sound, robust, and well described. A few minor concerns are listed below:

Comment #1: In multivariable model (Table 2) “cardiovascular diseases (incl. hypertension)” and “hypertension” were both included in the model; collinearity has been investigated in this case?

Response #1: We thank the reviewer for reading our manuscript and we appreciate comments and suggestions for improvement. The reviewers expressed concerns with regards to collinearity between “cardiovascular diseases (incl. hypertension)” and “hypertension”. 

First of all, we would like to explain why we included both variables in the model. We pre-selected relevant risk factors for the model based on literature and clinical relevance, rather than purely on the basis of statistical significance. Several types of “cardiovascular diseases” have been associated with increased risk of COVID-19 mortality, which is why this category was of interest. However, “cardiovascular diseases” is a very broad category and many studies and doctors have expressed interest in the effect of hypertension specifically. Therefore, we considered both variables separately to be relevant in our model. We have clarified this in our methods section, line 130-132: 

“These risk factors were pre-selected based on literature and expert opinion to be clinically relevant and routinely available at time of presentation, rather than based on statistical significance”. 

Nonetheless, the reviewer’s concern about collinearity remains valid. To show the effect of potential collinearity between these two variables on our findings, we have performed a sensitivity analysis, leaving out the variable ‘hypertension’. The table below reports the cause-specific hazard ratios of a multivariable model where “hypertension” is left out of the analysis. When comparing these results to the results from the model including hypertension, we see that the that the estimates of all variables are stable. In addition, the standard errors remain stable and the significance has not changed. Therefore, we can conclude that including “hypertension” as a separate variable has not influenced the reliability of our estimates. 

We have included these arguments in the discussion section, line 314-317:

“We included both cardiovascular disease and hypertension in our model, which may be subject to collinearity. However, a sensitivity analyses excluding hypertension from the model did not change our estimates and standard errors, showing that our reported estimates are valid.”

 Multivariable including hypertension Multivariable excluding hypertension

 Death Recovery Death Recovery

Age (years) 1.10 (1.08-1.12) * 0.99 (0.98-0.99) * 1.10 (1.08-1.12) * 0.99 (0.98-0.99) *

Sex, male 1.07 (0.79-1.47) 0.90 (0.75-1.08) 1.11 (0.82-1.51) 0.89 (0.74-1.07)

BMI 1.01 (0.98-1.04) 1.01 (0.99-1.03) 1.01 (0.97-1.04) 1.01 (0.99-1.03)

Diabetes Mellitus 1.17 (0.86-1.59) 0.85 (0.69-1.04) 1.14 (0.84-1.56) 0.86 (0.70-1.05)

Cardiovascular disease (incl. hypertension) 1.05 (0.69-1.59) 0.71 (0.54-0.93) * 0.90 (0.63-1.29) 0.78 (0.62-0.97) *

Hypertension 0.78 (0.56-1.10) 1.15 (0.90-1.48) 

Pulmonary disease 1.33 (0.98-1.80) 0.88 (0.73-1.07) 1.30 (0.96-1.75) 0.89 (0.74-1.08)

Immunocompromised a 1.46 (1.08-1.98) * 0.76 (0.62-0.93) * 1.50 (1.11-2.02) * 0.75 (0.62-0.92) *

Use of anticoagulant or antiplatelet medication 1.38 (1.01-1.88) * 1.15 (0.93-1.43) 1.41 (1.03-1.92) * 1.13 (0.91-1.39)

Use of ACE inhibitors and/or angiotensin II receptor blockers 0.99 (0.74-1.32) 1.09 (0.88-1.33) 0.95 (0.72-1.27) 1.11 (0.90-1.36)

Chest X-Ray 

 Performed, not suggestive for COVID-19

 Performed, suggestive for COVID-19

 Not performed 

Ref

1.07 (0.63-1.80) 

0.90 (0.54-1.48) 

Ref

1.52 (1.05-2.19) *

1.18 (0.82-1.70) 

Ref

1.07 (0.63-1.81)

0.90 (0.54-1.48) 

Ref

1.51 (1.05-2.18) *

1.18 (0.82-1.69)

CT scan severity score 1.01 (0.98-1.05) 0.97 (0.95-0.99) * 1.01 (0.98-1.05) 0.97 (0.95-0.99) *

Symptom duration (days) 0.98 (0.96-1.01) 1.01 (1.00-1.02) * 0.98 (0.96-1.01) 1.01 (1.01-1.02) *

Symptoms, fever 0.70 (0.52-0.95) * 1.05 (0.85-1.30) 0.69 (0.51-0.93) * 1.05 (0.85-1.30)

Symptoms, dyspnea 0.77 (0.58-1.03) 1.01 (0.84-1.21) 0.78 (0.58-1.04) 1.01 (0.84-1.21)

Modified Early Warning Score (MEWS) b 1.09 (1.01-1.18) * 0.97 (0.93-1.02) 1.08 (1.00-1.17) * 0.97 (0.93-1.02)

Neutrophil-to-lymphocyte rate c 0.97 (0.59-1.60) 1.01 (0.74-1.38) 1.00 (0.61-1.65) 1.00 (0.74-1.36)

Lactate dehydrogenase (U/L) c 6.68 (1.95-22.8) * 0.25 (0.13-0.48) * 6.67 (1.95-22.86) * 0.25 (0.13-0.49) *

Creatinine (µmol/L) c 1.84 (0.87-3.90) 1.17 (0.69-1.99) 1.78 (0.85-3.75) 1.21 (0.71-2.05)

Procalcitonin (µg/L) c 1.04 (0.76-1.41) 0.88 (0.75-1.03) 1.03 (0.76-1.39) 0.88 (0.75-1.03)

C-reactive protein (mg/L) c 1.25 (0.79-2.00) 0.88 (0.69-1.11) 1.22 (0.77-1.94) 0.89 (0.70-1.12)

Ferritin (µg/L) c 0.66 (0.43-1.02) 0.77 (0.60-0.99) * 0.66 (0.43-1.02) 0.78 (0.61-0.99) *

D-dimer (ng/L) d 0.99 (0.84-1.16) 0.94 (0.87-1.00) 0.99 (0.84-1.15) 0.94 (0.88-1.00) 

Comment #2 “Use of anticoagulants or antiplatelet medication” resulted connected with an increased risk of death. According to Authors hypothesis this association could be explained by the underlying cardiovascular disease. 

However: i) in multivariable model cardiovascular disease lost the statistically significance, implying that the use of anticoagulants/antiplatelets was independently associated with mortality; 

ii) a considerable risk of thromboembolic events was reported in course of COVID-19 (Bavaro DF, et al. Occurrence of Acute Pulmonary Embolism in COVID-19-A case series. Int J Infect Dis. 2020;98:225-226.).

In my opinion, this work (or similar ones) should be cited, and the risk of death according to use of anticoagulants should be better discussed since current guidelines suggest the use of low-molecular-weight-heparin in all hospitalized COVID-19 patients, if not contraindicated.

Response 2i and 2ii: we appreciate this valuable comment and we agree that the effect of “anticoagulants or antiplatelet medication” cannot be explained solely by the effect of cardiovascular diseases, but that it was also an independent risk factor in our results. We recognize that there is conflicting evidence with regards to the effect of these type of medications on COVID-19 mortality. We have elaborated on this issue in more detail in the discussion section of the revised manuscript, also including the reference the reviewer has suggested.

Discussion, line 323-333: 

“Although this may be partially explained by the fact that these medications are used by patients with cardiovascular disease, which has been reported as an individual risk factor for in-hospital mortality [4], anticoagulant and/or antiplatelet medication remained an independent risk factor for death in our multivariable analyses. Thromboembolic events are frequently reported in association with severe COVID-19 disease and mortality [28, 29], and current guidelines suggest prophylactic anticoagulants in all hospitalized COVID-19 patients if not contraindicated. However, studies have reported conflicting results regarding the effect of anticoagulants/antiplatelet medication on COVID-19 mortality [30], ranging from a protective effect [31] to a harmful effect [32, 33], or no association [31, 34]. Prospective studies and RCTs are needed to explore the true effects of these medications in hospitalized COVID-19 patients.”

---

## [Decision Letter · Decision Letter 1]

15 Mar 2021

Risk factors for in-hospital mortality in laboratory-confirmed COVID-19 patients in the Netherlands: a competing risk survival analysis.

PONE-D-20-40238R1

Dear Dr. Josephine S. van de Maat,

We’re pleased to inform you that your manuscript has been judged scientifically suitable for publication and will be formally accepted for publication once it meets all outstanding technical requirements.

Kind regards,

Francesco Di Gennaro

Academic Editor

PLOS ONE

Additional Editor Comments (optional):

dear authors congratulations

Reviewers' comments:

Reviewer's Responses to Questions

**Comments to the Author**

1. If the authors have adequately addressed your comments raised in a previous round of review and you feel that this manuscript is now acceptable for publication, you may indicate that here to bypass the “Comments to the Author” section, enter your conflict of interest statement in the “Confidential to Editor” section, and submit your "Accept" recommendation.

Reviewer #1: All comments have been addressed

Reviewer #2: All comments have been addressed

2. Is the manuscript technically sound, and do the data support the conclusions?

Reviewer #1: Yes

Reviewer #2: Yes

3. Has the statistical analysis been performed appropriately and rigorously? 

Reviewer #1: Yes

Reviewer #2: Yes

4. Have the authors made all data underlying the findings in their manuscript fully available?

Reviewer #1: Yes

Reviewer #2: Yes

5. Is the manuscript presented in an intelligible fashion and written in standard English?

Reviewer #1: Yes

Reviewer #2: Yes

6. Review Comments to the Author

Reviewer #1: thank you. conviced and satisfied with your response to the comments. the data are helpful and may be usful for stratification of the patients

Reviewer #2: (No Response)

7. PLOS authors have the option to publish the peer review history of their article (what does this mean?). If published, this will include your full peer review and any attached files.

Reviewer #1: **Yes: **Aly Ahmed Abdel Rahim

Reviewer #2: No

---

## [Editor Report · Acceptance letter]

18 Mar 2021

PONE-D-20-40238R1 

Risk factors for in-hospital mortality in laboratory-confirmed COVID-19 patients in the Netherlands: a competing risk survival analysis. 

Dear Dr. van de Maat:

I'm pleased to inform you that your manuscript has been deemed suitable for publication in PLOS ONE. Congratulations! Your manuscript is now with our production department. 

Kind regards, 

on behalf of

Dr. Francesco Di Gennaro 

Academic Editor

PLOS ONE